# Prevalence and factors associated with psychological distress among key populations in Togo, 2017

Martin Kouame Tchankoni[1]*, Fifonsi Adjidossi Gbeasor-Komlanvi[1,2], Alexandra Marie Bitty-Anderson[3], Essèboè Koffitsè Sewu[1], Wendpouiré Ida Carine Zida-Compaore[1], Ahmadou Alioum[4,5], Mounerou Salou[6], Claver Anoumou Dagnra[6], Didier Koumavi Ekouevi[1,2,3,4,5]

1 African Center for Epidemiology and Public Health Research (CARESP), Lomé, Togo, 2 Department of Public Health, University of Lomé, Faculty of Health Sciences, Lomé, Togo, 3 PACCI Program—ANRS Ivory Coast Site, Treichville University Hospital, Abidjan, Ivory Coast, 4 University of Bordeaux, Inserm Center, Bordeaux, France, 5 University of Bordeaux, Institute of Public Health Epidemiology Development (ISPED), Bordeaux, France, 6 Department of Basic Sciences, University of Lomé, Center for Molecular Biology and Immunology, Lomé, Togo

* tchankonimartin@yahoo.fr

**Data Availability Statement:** All relevant data are within the manuscript and its Supporting Information files.

## Abstract

### Objectives

Mental health is a largely neglected issue among in Sub-Saharan Africa, especially among key populations at risk for HIV. The aim of this study was to estimate the prevalence of psychological distress (PD) and to assess the factors associated among males who have sex with males (MSM), female sex workers (FSW) and drug users (DU) in Togo in 2017.

### Study design

A cross-sectional bio-behavioral study was conducted in August and September 2017 using a respondent-driven sampling (RDS) method, in eight cities in Togo.

### Methods

A standardized questionnaire was used to record sociodemographic characteristics and sexual behaviors. The Alcohol Use Disorders Identification Test (AUDIT) and a subset of questions from the Tobacco Questions for Survey were used to assess alcohol and tobacco consumption respectively. PD was assessed with the Kessler Psychological Distress Scale. A blood sample was taken to test for HIV. Descriptive statistics, univariable and multivariable ordinal regression models were used for analysis.

### Results

A total of 2044 key populations including 449 DU, 952 FSW and 643 MSM with a median age of 25 years, interquartile range (IQR) [21–32] were recruited. The overall prevalence of mild PD among the three populations was 19.9% (95%CI = [18.3–21.8]) and was 19.2% (95%CI = [17.5–20.9]) for severe/moderate PD. HIV prevalence was 13.7% (95%CI =

**Funding:** This work was supported by the « Centre Africain de Recherche en Epidemiologie et en Santé Publique (CARESP) » (African Center for Epidemiology and Public Health Research) and the Togo National HIV/AIDS and STI Control Program. The funders had no role in study design, data collection and analysis, decision to publish, or preparation of the manuscript.

**Competing interests:** The authors have declared that no competing interests exist.

**Abbreviations:** 95% CI, 95% confidence interval; aOR, adjusted Odds Ratio; IQR, interquartile range; PD, psychological distress.

[12.2–15.2]). High age ($\geq$ 25 years) [aOR = 1.24 (95% CI: 1.02–1.50)], being HIV positive [aOR = 1.80 (95% CI: 1.31–2.48)] and hazardous alcohol consumption [aOR = 1.52 (95% CI: 1.22–1.87)] were risk factors for PD. Secondary [aOR = 0.52 (95% CI: 0.42–0.64)] or higher [aOR = 0.46 (95% CI: 0.32–0.64)] education levels were protective factors associated with PD. FSW [OR = 0.55 (95% CI: 0.43–0.68)] and MSM [OR = 0.33 (95% CI: 0.24–0.44)] were less likely to report PD compared with DU.

## Conclusion and recommendations

This is the first study conducted among a large, nationally representative sample of key populations in Togo. The prevalence of PD is high among these populations in Togo and was associated to HIV infection. The present study indicates that mental health care must be integrated within health programs in Togo with a special focus to key populations through interventions such as social support groups.

## Introduction

Mental health disorders represent a growing public health challenge worldwide. According to the World Health Organization (WHO), mental health is a state of well-being in which every individual realizes his or her own potential, can cope with the normal stresses of life, can work productively and fruitfully, and is able to make a contribution to her or his community [1]. Between 1990 and 2010, the burden of mental health disorders, such as depression and other common mental disorders, alcohol-use/substance-use disorders, and psychoses, increased by 37.6% [2,3]. Overall, mental health disorders are key contributors to nearly 10 to 14% of the global burden of disease, including death and disability [3,4]. Several factors have been associated with mental health, including gender, marital status, education, tobacco use, partner control/abuse [5,6].

Some population subgroups, such as those at higher risk of HIV are particularly burdened with mental health issues. According to the UNAIDS, gay men and other men who have sex with men (MSM), female sex workers (FSW) and their clients, transgender people, people who inject drugs and prisoners and other incarcerated people are the main key population groups [7]. In a study conducted in the Netherlands in 1999, mental health disorders were more prevalent among homosexually active people compared with their heterosexually active counterparts [8]. In addition, Luo et al. in a meta-analysis, reported that the pooled lifetime prevalence of suicidal ideation among MSM was 34.9% [9]. Similarly, using the Center for Epidemiological Studies Short Depression Scale, FSW in the Dominican Republic were more frequently screened positive for depression compared with HIV negative women who were not FSW (70.2% vs 52.2%) [10]. Moreover, poor mental health has been identified as a key contributor to the HIV epidemic among key populations. Indeed, MSM who are mentally depressed engage in unprotected sex and substance abuse [11–13]. Several studies have demonstrated a significant association between depressive symptoms and HIV and other sexually transmitted infections (STI) among MSM [14–16]. Poor mental health is a major concern in sub-Saharan Africa which bears the heaviest burden of HIV/AIDS [17]. As an integral part of overall well-being, mental health is not on the front line in terms of priority for health practitioners and it has been a largely neglected issue among marginalized populations in the developing world [17,18].

The high rates of HIV/AIDS among MSM and other key populations have pushed preventive interventions to focus on the risks associated with HIV/AIDS transmission and this situation has led to mental health neglect [19]. In Togo, HIV prevalence among key populations ranges from 11% to 13% compared to 2.1% in the general population [20,21]. Although issues related to key populations have been extensively studied in Togo, to our knowledge, no data are available on mental health problems among these populations. The aim of this study was to assess the prevalence of and factors associated with PD among MSM, FSW and DU in Togo in 2017.

## Methods

### Study design and sampling

This study was a bio-behavioral cross-sectional study conducted from August to September 2017 in eight cities in Togo. Prior to the study, locations (associations and hot spots) specific to each group of key population were identified during preliminary visits with the help of leaders from these communities. DU and FSW were recruited in drug-dealing/consumption locations and brothels (licensed or not), respectively. MSM were recruited using a Respondent Driven Sampling (RDS) method. MSM community leaders were the first "seeds". A total of 28 seeds were identified at first based on their roles in their community and on their representativeness. Each seed from the first wave selected had to represent at least one sub-group: the actives (the ones who penetrate during sex), the passives (the penetrated), bisexuals, gays, etc. Each participant was then given three coupons with a unique identification code to recruit three other seeds in their network until the required sample size for each group was reached. Inclusion criteria for the three groups were being 18 years or older, living/working/studying in Togo for a minimum of 3 months at the time of the study, and being in possession of a recruitment coupon. In addition to these criteria, criteria specific to MSM were having had anal sex with a man in the previous 12 months, for FSW having had sex in exchange for money as a compensation in the previous 12 months and for DU, consuming heroin, cocaine or hashish at the time of the study.

### Ethics approval and consent to participate

This study was approved by the "Comité de Bioéthique pour la Recherche en Santé" (Bioethics Committee for Health Research) from the Togo Ministry of Health (CBRS No. 18/2017/CBRS of June 22nd 2017). Potential participants were informed about the study purpose and procedures, potential risks and protections, and compensation. Informed consent was documented with signed consent forms prior to participation.

### Data collection

After eligibility screening and written informed consent, a structured and standardized questionnaire was administered by trained study staff during a face-to-face interview. The Kessler PD Scale (K10) [22] was used to measure PD. The K10 is a 10-item screening tool which has been used in several countries and it has been validated in a low-income, African setting with strong validity and reliability [23,24].

Alcohol and tobacco consumption were also assessed using validated tools: the Alcohol Use Disorders Identification Test (AUDIT) [25] and a subset of the Tobacco Questions for Surveys [26] respectively. To collect information on socio-demographic characteristics, risky sexual behaviors, STIs, HIV prevention methods, HIV testing history, access to health care services, and HIV knowledge, items from the Family Health International (FHI) 360 validated guide for

bio-behavioral surveys [27] were included in the questionnaire. The same questionnaires were used across the three populations with slight adaptations depending on the population.

## Laboratory testing

Two HIV rapid tests were performed using SD Biolane Duo VIH/Syphilis rapid test kits. For each participant, a 4 mL blood sample was collected and a third test, ImmunoComb II VIH1-2 Comfirm (Orgenics Ltd, Israël) was used for confirmation. All biological test analyses were completed in the principal HIV laboratory research unit, the Molecular Biology Laboratory (CBMI) at the University of Lomé.

## Scores and operational definitions

**Psychological distress.** The K10 has been examined and validated among several populations and aims at measuring anxiety and depression with a 10-item questionnaire, each question pertaining to an emotional state and a five-level response scale for each response. The score obtained from the scale allows categorization of participants into four ordinal categories of PD: severe (score $\geq$ 30), moderate (score: 25–29), mild (score: 20–24) and none (score < 20) [28]. In our study, due to the small number of participants falling in the severe category, we collapsed the Kessler PD scores into three categories as: severe/moderate (score $\geq$25), mild (score: 20–24) and none (score <20).

**Alcohol consumption.** To assess alcohol consumption, the AUDIT-C was used. The AUDIT-C is a screening tool used to identify persons who are hazardous drinkers or have active alcohol use disorders (including alcohol abuse or dependence). The AUDIT-C is a modified version of the 10 question AUDIT instrument and it proved to be as effective as the AUDIT [29]. As with the 10-item AUDIT, higher scores indicate more problematic alcohol use. The AUDIT-C is scored on a scale of 0–12. Each question has 5 answer choices and can obtain a score from 0 to 4. A score $\geq$5 for men and $\geq$4 for women indicates hazardous drinking, while a score of 0 indicates a non-drinker; moderate alcohol use lies in-between [25,30,31].

## Statistical analysis

We used a Partial Proportional Odds (PPO) model to assess factors associated to PD. Descriptive statistics were performed and results were presented with frequency tabulations and percentages. Prevalence rates were estimated with their 95% confidence interval. For model building, characteristics that had a p-value <0.20 in univariable analysis were considered for the full multivariable models, which were subsequently finalized using a stepwise, backward elimination approach (p-value <0.05). Predictor variables were selected as those found to be relevant according to the literature review. All computations were conducted using SAS version 9.4 (SAS Institute Inc., Cary, NC, USA).

## Results

### Sociodemographic and clinical characteristics

A total of 2044 key populations including 449 DU, 952 FSW and 643 MSM with a median age of 25 years, interquartile range (IQR) [21–32 years] were included. More than two-thirds (68.5%) of participants had at least a secondary education level and 15.7% were married or living with a partner. The HIV prevalence was 13.7% (95%CI = [12.2–15.2]) across the three key populations with the highest prevalence among MSM (21.6%; 95%CI = [18.5–25.0]). Other sociodemographic and clinical characteristics are summarized in Table 1.

**Table 1. Sociodemographic and clinical characteristics of key populations, Togo 2017.**

|  | DU (n = 449) | FSW (n = 952) | MSM (n = 643) | Total (= 2044) |
|---|---|---|---|---|
| **Age (years), median [IQR]** | 29 [23–37] | 26 [22–32] | 23 [20–26] | 25 [21–32] |
| **Marital status, n (%)** |  |  |  |  |
| Married/Living with a partner | 148 (33.0) | 133 (14.0) | 40 (6.2) | 321 (15.7) |
| Not married | 301 (67.0) | 819 (86.0) | 603 (93.8) | 1723 (84.3) |
| **Education level, n (%)** |  |  |  |  |
| Primary school | 160 (35.6) | 429 (45.1) | 54 (8.4) | 643 (31.5) |
| Secondary school | 268 (59.7) | 476 (50.0) | 358 (55.7) | 1102 (53.9) |
| College/University | 21 (4.7) | 47 (4.9) | 231 (35.9) | 299 (14.6) |
| **Religion, n (%)** |  |  |  |  |
| Other/ Non-believer | 97 (21.6) | 110 (11.5) | 65 (10.1) | 272 (13.2) |
| Christian | 282 (62.8) | 747 (78.5) | 525 (81.7) | 1554 (76.0) |
| Muslim | 70 (15.6) | 95 (10.0) | 53 (8.2) | 220 (10.8) |
| **Alcohol consumption, n (%)** |  |  |  |  |
| Non-drinker | 66 (14.7) | 323 (33.9) | 326 (50.7) | 715 (34.9) |
| Moderate drinking | 79 (17.6) | 197 (20.7) | 130 (20.2) | 406 (19.9) |
| Hazardous consumption | 304 (67.7) | 432 (45.4) | 187 (29.1) | 923 (45.2) |
| **Tobacco consumption, n (%)** |  |  |  |  |
| Non-smoker | 89 (19.8) | 825 (86.7) | 546 (84.9) | 1460 (71.4) |
| Smoker | 360 (80.2) | 127 (13.3) | 97 (15.1) | 584 (28.6) |
| **HIV infection, n (%)** |  |  |  |  |
| No | 433 (96.4) | 827 (86.9) | 504 (78.4) | 1764 (86.3) |
| Yes | 16 (3.6) | 125 (13.1) | 139 (21.6) | 280 (13.7) |

IQR: interquartile range

DU: drug user; FSW: female sex worker; MSM: men who have sex with men

## Prevalence of psychological distress

The median Kessler score was 14 (IQR = [10–19]) for MSM, 18 (IQR = [13–23]) for FSW and 21 (IQR = [16–27]) for DU. About two-thirds (n = 1,244; 61.0%) of the participants did not have PD. The overall prevalence of mild PD among the three populations was 19.9% (95%CI = [18.3–21.8]) and was 19.2% (95%CI = [17.5–20.9]) for the severe/moderate PD. The prevalence of PD was statistically different in the three groups (p-value <0.001). Among DU, the prevalence of mild PD was 23.4%, 95%CI = [20–28] and 32.1%, 95%CI = [28–37] had severe/moderate PD. Severe/moderate PD was reported among one FSW in five (19.0%; 95%CI = [17–22]). The relationships between sociodemographic and clinical characteristics and PD are presented in Table 2.

## Factors associated with psychological distress

Results of ordinal logistic regression models among key population are presented in Table 3.

In univariable analysis, age, marital status, education level, religion, alcohol consumption, tobacco consumption, HIV infection status and the group of key population were associated with PD (Table 3).

In multivariable analysis, after adjustment on the other variables, respondents who were 25 years old and older, were more likely to have severe/moderate or mild PD than respondents who were younger (p = 0.026). HIV positive serological status was a risk factor for PD (aOR = 1.80; 95%CI [1.31–2.48]). The odds of PD were higher among hazardous drinkers

**Table 2. Associations between psychological distress[a] and sociodemographic and clinical characteristics among key populations (N = 2044).**

| | Psychological distress | | | |
| --- | --- | --- | --- | --- |
| | **No** | **Mild** | **Severe/Moderate** | **P-value** |
| **Prevalence, n (%)** | 1244 (60.9) | 408 (19.9) | 392 (19.2) | |
| **Age (years), median [IQR]** | 24 [21–30] | 26 [21–33] | 28 [22.5–34] | **<0.001**[*] |
| **Marital status, n (%)** | | | | **<0.001**[**] |
| Not married | 1081 (62.7) | 334 (19.4) | 308 (17.9) | |
| Married/Living with a partner | 163 (50.8) | 74 (23.0) | 84 (26.2) | |
| **Education level, n (%)** | | | | **<0.001**[**] |
| Primary school | 292 (45.4) | 164 (25.5) | 187 (29.1) | |
| Secondary school | 725 (65.8) | 208 (18.9) | 169 (15.3) | |
| College/University | 227 (76.0) | 36 (12.0) | 36 (12.0) | |
| **Religion, n (%)** | | | | **0.003**[**] |
| Other/ Non-believer | 136 (50.0) | 67 (24.6) | 69 (25.4) | |
| Christian | 973 (62.6) | 297 (19.1) | 284 (18.3) | |
| Muslim | 135 (61.9) | 44 (20.2) | 39 (17.9) | |
| **Alcohol consumption, n (%)** | | | | **<0.001**[****] |
| Non-drinker | 502 (70.2) | 117 (16.4) | 96 (13.4) | |
| Moderate drinking | 244 (60.1) | 83 (20.4) | 79 (19.5) | |
| Hazardous consumption | 498 (54.0) | 208 (22.5) | 217 (23.5) | |
| **Tobacco consumption, n (%)** | | | | **<0.001**[**] |
| Non-smoker | 1069 (63.3) | 333 (19.7) | 288 (17.0) | |
| Smoker | 175 (49.4) | 75 (21.2) | 104 (29.4) | |
| **HIV infection, n (%)** | | | | **0.010**[**] |
| No | 1075 (60.9) | 366 (20.8) | 323 (18.3) | |
| Yes | 169 (60.4) | 42 (15.0) | 69 (24.6) | |
| **Key population** | | | | **<0.001**[**] |
| DU | 200 (44.5) | 105 (23.4) | 144 (32.1) | |
| FSW | 548 (57.6) | 223 (23.4) | 181 (19.0) | |
| MSM | 496 (77.1) | 80 (12.5) | 67 (10.4) | |

[a] Psychological distress was measured with the Kessler Psychological Distress Scale (10). Total Kessler score <20: No Psychological distress, 20–24: Mild disorder, 25–50: Moderate/Severe disorder.

[*] Kruskall-Wallis test

[**] Chi-square test

IQR: interquartile range

DU: drug user; FSW: female sex worker; MSM: men who have sex with me

(aOR = 1.52; 95%CI = [1.22–1.87]) compared to non-drinkers. Those with secondary school level or with college/university level were less likely to have severe/moderate or mild PD compared to those who stopped in primary school (p<0.001). Compared to DU, FSW (aOR = 0.55; 95%CI = [0.43–0.68]) and MSM (aOR = 0.33; 95%CI = [0.24–0.44]) were less likely to have severe/moderate or mild PD.

## Discussion

Among MSM, FSW, and DU in Togo, the overall prevalence of mild and severe/moderate psychological distress was 19.9% and 19.2% respectively. DU had higher level of psychological distress compared with FSW and MSM (p<0.001). Also, among the three populations, participants with HIV infection were more likely to experience psychological distress than

**Table 3. Factors associated with psychological distress among key population in Togo (Partial proportional odds model[#]).**

| | (Psychological distress: 1 = Severe/Moderate, 2 = Mild, 3 = No) | | | | | |
|---|---|---|---|---|---|---|
| | Univariable | | | Multivariable | | |
| | 1 vs (2 and 3) | (1 and 2) vs 3 | P-value | 1 vs (2 and 3) | (1 and 2) vs 3 | P-value |
| | OR (95% CI) | OR (95% CI) | | aOR (95% CI) | aOR (95% CI) | |
| **Age (years)** | | | | | | |
| < 25 | Reference | | <0.001 | Reference | | 0.026 |
| ≥ 25 | 1.77 (1.48–2.11) | 1.77 (1.48–2.11) | | 1.24 (1.02–1.50) | 1.24 (1.02–1.50) | |
| **Marital status** | | | <0.001 | | | - |
| Not married | Reference | | | - | - | |
| Married/Living with a partner | 1.63 (1.29–2.04) | 1.63 (1.29–2.04) | | - | - | |
| **Education level** | | | <0.001 | | | <0.001 |
| Primary school | Reference | | | Reference | | |
| Secondary | 0.44 (0.36–0.52) | 0.44 (0.36–0.52) | | 0.52 (0.42–0.64) | 0.52 (0.42–0.64) | |
| College/University | 0.27 (0.20–0.37) | 0.27 (0.20–0.37) | | 0.46 (0.32–0.64) | 0.46 (0.32–0.64) | |
| **Religion** | | | <0.001 | | | - |
| Other/ Non-believer | Reference | | | - | - | |
| Christian | 0.62 (0.48–0.78) | 0.62 (0.48–0.78) | | - | - | |
| Muslim | 0.63 (0.44–0.88) | 0.63 (0.44–0.88) | | - | - | |
| **Alcohol consumption** | | | <0.001 | | | <0.001 |
| Non-drinker | Reference | | | Reference | | |
| Moderate drinking | 1.56 (1.22–2.0) | 1.56 (1.22–2.0) | | 1.28 (0.98–1.65) | 1.28 (0.98–1.65) | |
| Hazardous consumption | 2.0 (1.64–2.45) | 2.0 (1.64–2.45) | | 1.52 (1.22–1.87) | 1.52 (1.22–1.87) | |
| **Tobacco consumption** | | | <0.001 | | | - |
| Non-smoker | Reference | | | | | |
| Smoker | 1.84 (1.46–2.32) | 1.48 (1.22–1.80) | | - | - | |
| **HIV infection** | | | 0.004 | | | <0.001 |
| No | Reference | | | Reference | | |
| Yes | 1.46 (1.08–1.96) | 1.02 (0.79–1.32) | | 1.80 (1.31–2.48) | 1.25 (0.94–1.65) | |
| **Key population** | | | <0.001 | | | <0.001 |
| DU | Reference | | | Reference | | |
| FSW | 0.56 (0.45–0.69) | 0.56 (0.45–0.69) | | 0.55 (0.43–0.68) | 0.55 (0.43–0.68) | |
| MSM | 0.23 (0.18–0.30) | 0.23 (0.18–0.30) | | 0.33 (0.24–0.44) | 0.33 (0.24–0.44) | |

[#] For nearly all variables, the left 2 columns are the same, so are the right 2 columns, because the assumption of proportional odds was met for those variables and therefore we used the same value for both columns. (Only the OR for tobacco smoker and HIV infection were different.)

their non-infected counterparts. Factors associated with psychological distress other than HIV infection were greater age, lower education level and hazardous alcohol consumption.

We did not have national data or data among key populations in Togo to which we could compare our findings. Other studies using K-10 were conducted in Africa. In the general population in South Africa (n = 25850), 23.9% of study participants reported psychological distress (K-10≥20) [34]. Hazardous alcohol drinking (OR = 1.79) as well as HIV positive status (OR = 4.76) were associated with psychological distress as observed in our study. In another cross-sectional study among 180 patients conducted in drug rehabilitation centers in Nepal which used K-6 as psychological distress measurement tool, the prevalence of high psychological distress was 51.1% [35]. Factors associated with psychological distress were age, education level, severity of drug abuse. Among Latino and African American MSM living in the USA,

22% had moderate/severe psychological distress using the K-10 scale [36]. This prevalence is similar to the overall moderate/severe psychological distress prevalence in our study among the three key populations (19.0%). However, this estimate was higher than that reported among MSM in our study (10.4%). There is a gap between the percentages of PD in other countries and in our study mainly because of the age differences. Indeed, the population considered in our study is relatively young (median age = 25 years) unlike in other studies [35,36]. Also, different tools have been used in the other studies to measure PD as the Patient Health Questionnaire (PHQ) [19], the Center for Epidemiologic Studies Depression Scale (CES-D) [40] and K-6 [35].

Several studies in developed countries have reported that alcohol or drug consumption are closely associated with psychological distress. This is consistent with our study regarding alcohol consumption. Binge drinking or cocaine consumption were associated with psychological distress among Latino and African American in the USA [36].

Among 280 HIV infected patients in our study, 39.6% had psychological distress (K-10 scale>20). In a Nigerian study among 117 people living with HIV/AIDS recruited in a teaching hospital, 47.9% participants had psychological distress (K-10 scale>20) [37] and alcohol use was associated with PD. Similar results were reported in Ethiopia where an association between PD and alcohol use disorder was reported (OR = 1.90) [38].

A limited number of studies among key populations used the Kessler scale. However mental health was explored in these populations with tools such as the PHQ and the CES-D. A study in India among MSM (n = 1176) measured the prevalence of depression with PHQ-2 tool. Having had an STI in the six months preceding the study, being HIV positive, not having used a condom during sexual intercourse were associated with depression [19]. In the USA, a cross-sectional survey was conducted among Lesbian, Gay male, and Bisexual (LGB) older adults, aged 50 and older (n = 2439). This study investigated depression among LGB with the CES-D tool and the findings revealed that lifestyle, financial barriers, obesity were factors which accounted for poor general health, and depression among LGB older adults [39]. Another study conducted in the USA among black MSM (n = 197) used the CES-D to measure depression. This study has also reported a link between alcohol consumption and depressive symptoms [40].

Little is known about mental health among key populations in sub-Saharan Africa, and to our knowledge this study is the first one which has assessed psychological distress among these groups using an international validated tool (K-10). This tool has been validated in the African context in South Africa in general population [34]. In addition, we used data from a nationally representative study (n = 2044) including three main groups of key populations, allowing for the generalization of our findings. Finally, our study provided useful information on factors associated with PD among key populations in order to design specific interventions within these populations.

Our study has some limitations. First, we worked on three populations, but it is possible that an FSW may be a DU or an MSM may also be a DU. However, to classify the population, we took into account only the entry point. Since our study was based on self-reporting, social desirability bias cannot be ruled out. This bias could have underestimated the prevalence rates of PD. Also, as we conducted a cross-sectional study, causality between the factors identified and PD could not be addressed. Other factors such as an employment status were not included although they could potentially affect psychological distress, as reported in a study conducted among MSM in South Africa [34]. Finally, there was no medical evaluation in the present study.

## Conclusion and recommendations

This is the first study conducted among a large, nationally representative sample of key populations in Togo. The prevalence of PD is high among Key Population in Togo and was associated to HIV infection. Mental health care such as social support groups must be integrated within health and prevention programs dedicated to key populations in Togo in order to promote a holistic perspective of their health. In order to confirm the high prevalence of PD, studies in the general population or comparative studies are needed. Also, since PD a multifactorial process, it would be useful to couple a qualitative study with a quantitative study.

## Supporting information

**S1 Data.**
(XLSX)

**S2 Data.**
(XLSX)

**S3 Data.**
(XLSX)

## Acknowledgments

We are thankful to the key populations who accepted to participate in this study and to the final year medical students of the 'Faculté des Sciences de la Santé-Université de Lomé' who performed data collection for the study.

## Author Contributions

**Conceptualization:** Fifonsi Adjidossi Gbeasor-Komlanvi, Alexandra Marie Bitty-Anderson, Mounerou Salou, Claver Anoumou Dagnra, Didier Koumavi Ekouevi.

**Data curation:** Martin Kouame Tchankoni, Alexandra Marie Bitty-Anderson, Essèboè Koffitsè Sewu, Wendpouiré Ida Carine Zida-Compaore.

**Formal analysis:** Martin Kouame Tchankoni, Essèboè Koffitsè Sewu, Ahmadou Alioum.

**Investigation:** Fifonsi Adjidossi Gbeasor-Komlanvi, Mounerou Salou.

**Methodology:** Martin Kouame Tchankoni, Ahmadou Alioum.

**Software:** Martin Kouame Tchankoni.

**Supervision:** Wendpouiré Ida Carine Zida-Compaore, Didier Koumavi Ekouevi.

**Validation:** Ahmadou Alioum, Didier Koumavi Ekouevi.

**Writing – original draft:** Martin Kouame Tchankoni, Essèboè Koffitsè Sewu, Didier Koumavi Ekouevi.

**Writing – review & editing:** Fifonsi Adjidossi Gbeasor-Komlanvi, Alexandra Marie Bitty-Anderson, Wendpouiré Ida Carine Zida-Compaore, Ahmadou Alioum, Mounerou Salou, Claver Anoumou Dagnra.

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
