## [Decision Letter · Decision Letter 0]

13 Jan 2020

PONE-D-19-29894

Prevalence and factors associated with psychological distress among key populations in Togo, 2017.

PLOS ONE

Dear MPH TCHANKONI,

Thank you for submitting your manuscript to PLOS ONE. After careful consideration, we feel that it has merit but does not fully meet PLOS ONE’s publication criteria as it currently stands. Therefore, we invite you to submit a revised version of the manuscript that addresses the points raised during the review process.

We would appreciate receiving your revised manuscript by Feb 27 2020 11:59PM. To enhance the reproducibility of your results, we recommend that if applicable you deposit your laboratory protocols in protocols.io, where a protocol can be assigned its own identifier (DOI) such that it can be cited independently in the future. For instructions see: http://journals.plos.org/plosone/s/submission-guidelines#loc-laboratory-protocols

We look forward to receiving your revised manuscript.

Kind regards,

Joel Msafiri Francis, MD, MS, PhD

Academic Editor

PLOS ONE

Journal Requirements:

Please provide an amended Funding Statement that declares *all* the funding or sources of support received during this specific study (whether external or internal to your organization) as detailed online in our guide for authors at http://journals.plos.org/plosone/s/submit-now.

4. . PLOS requires an ORCID iD for the corresponding author in Editorial Manager on papers submitted after December 6th, 2016. Please ensure that you have an ORCID iD and that it is validated in Editorial Manager. To do this, go to ‘Update my Information’ (in the upper left-hand corner of the main menu), and click on the Fetch/Validate link next to the ORCID field. This will take you to the ORCID site and allow you to create a new iD or authenticate a pre-existing iD in Editorial Manager. Please see the following video for instructions on linking an ORCID iD to your Editorial Manager account: https://www.youtube.com/watch?v=_xcclfuvtxQ

5. Your ethics statement must appear in the Methods section of your manuscript. If your ethics statement is written in any section besides the Methods, please move it to the Methods section and delete it from any other section. Please also ensure that your ethics statement is included in your manuscript, as the ethics section of your online submission will not be published alongside your manuscript.

Reviewers' comments:

Reviewer's Responses to Questions

**Comments to the Author**

1. Is the manuscript technically sound, and do the data support the conclusions?

Reviewer #1: Yes

Reviewer #2: Yes

2. Has the statistical analysis been performed appropriately and rigorously? 

Reviewer #1: Yes

Reviewer #2: Yes

3. Have the authors made all data underlying the findings in their manuscript fully available?

Reviewer #1: Yes

Reviewer #2: Yes

4. Is the manuscript presented in an intelligible fashion and written in standard English?

Reviewer #1: Yes

Reviewer #2: Yes

5. Review Comments to the Author

Reviewer #1: Thanks for the opportunity to read and learn about your interesting work. May I offer the following suggestions.

General:

It is important to be clear about the direction of associations in key places like the Abstract and in the first paragraph of Discussion. I think most of what you say in the Results section of the Abstract is okay *except* regarding education. I recommend you end the list of factors with "hazardous alcohol consumption (aOR and 95% CI are listed here) were associated with PD." Then start a new sentence saying something like "People with secondary (aOR and 95% CI here) or higher (aOR and CI) education levels were LESS likely to report PD." Then in the first paragraph of Discussion section (page 11), you should specify GREATER age and LOWER education level.

Where you refer to bivariate and multivariate methods, my understanding is these should be labeled Univariable and Multivariable (ending with -able instead of -ate). See https://www.ncbi.nlm.nih.gov/pmc/articles/PMC3679183/ for discussion.

There are several places where you splice two sentences together with the word "and." Probably better to separate to 2 sentences, or use a semicolon instead of the "and." e.g., near bottom of page 2 after citations (11-13).

Specific:

Page 1 Abstract, last line in Methods section, should "model" have an -s on it ? Also, not sure if you can use abbreviation PD without introducing it, even though it appears later in the sentence in the name of the scale.

Regarding Results section of Abstract, see note below about so many zeros after decimal points. for median and IQR, delete the ".0" because you are using only integer values. For the percentages, some of the ".0" are actually not correct. 19.0% should be 19.2% (392/2044). Although I did not calculate the CI's, those too are not correct and you will have to calculate those yourself (where it says 18.0-22.0 and 17.0-21.0).

page 2, 2nd paragraph. I would change the 1st 2 sentences around a bit: "Some population subgroups, such as those at higher risk of HIV, are particularly burdened with mental health issues. According to UNAIDS (etc …)"

page 2, 8th line from bottom, insert "the" so it says "key contributor to the HIV epidemic"

page 3 line 7, add "of" so it says "prevalence of and factors associated with"

Need just a little more info on what "passives, actives" refers to - is it sexual position ?

Last paragraph on Kessler PD Scale - some of this text is repeated on the next page; delete one or the other.

Page 4 -- should be only one L in syphilis. Then 2 lines down, delete s on "tests"

next paragraph is where Kessler PD repeat occurs. Also in that paragraph, change "allows to categorize" to "allows categorization of."

page 5: along with "latent" mention that PO model is used for *ordinal* outcome variables.

With backward elimination approach, what was the criterion for elimination ? e.g., P<0.05, 0.01 ?

page 6: first sentence regarding Kessler scores, leave off all the zeros after decimal for median and IQR scores, since these Kessler score values are always integers. On the other hand, in the next sentence some of the zeros after decimal point in the percentages are not correct. 20.0% is correct, but 61.0% should be 60.9%, and 19.0% should be 19.2%. You will need to recalculate your 95% CI's in this sentence and the rest of this paragraph - these should not all be ending with ".0%"

I don't think you need Figure 1. Instead, just add the key population groups as another variable at the bottom of Table 2.

In Table 2, you need to make the same corrections to percentages (first row only) as in the text, namely change 61.0 to 60.9 and 19.0 to 19.2. On age, delete the ".0"s since all these numbers are integers only. Throughout the rest of Table 2, I think the ROW totals should add to 100% rather than the column totals. That helps the reader quickly approximate the odds ratios you will be presenting in Table 3.

Page 9: Change 1st sentence to: "RESULTS OF ordinal logistic regression models (etc...)" (add s to "models")

Line 2, here I would say univariable analysis instead of univariate. (and note you said "bivariate" in the abstract, which I think should also say "univariable")

Line 3, delete the word "factors" so it just says "were associated with …"

Next paragraph, change multivariate to multivariable.

Add an s so it says odds in "The odd of psychological" and then change "was" to "were"

Table 3: change column headers to Univariable and Multivariable. I think you should explain in a footnote that for nearly all variables, the left 2 columns are the same, and the right 2 columns are the same, because the assumption of proportional odds was met for those variables and therefore you used the same value for both columns. (only the tobacco smoker and HIV infection OR's are different.)

page 11, it is confusing to use "respectively" when there are 2 different lists. I suggest "Among MSM, FSW, and DU in Togo, the overall prevalence of mild and severe/moderate psychological distress was 20.0% and 19.2% respectively."

2nd paragraph, Could you say more about whether and why these percentages reporting PD in other countries are higher or lower than in your study ? In the sentence where you cite the 23.9% in South Africa, you could say "compared to 39.2% in our study."

Best wishes with the manuscript and your research.

Reviewer #2: Review for article titled: Prevalence and Factors Associated with Psychological Distress among Key Populations in Togo, 2017

Overall, the paper reads well, congratulations!

I have minor comments that I think need to be addressed:

Abstract

1. The write up has several grammatical errors that should be worked on.

2. Use of abbreviations should come after you have spelled the long form earlier. In the abstract, the author has used the abbreviation PD for the first time before it is written in full.

3. In reporting factors associated with PD, the author could choose to report such factors in groups; that is those that increase the risk and those that reduce the risk.

4. The conclusion sounds more like a recommendation. I advise that you give concluding statements based on your findings. You may want to title the section as Conclusion and Recommendation and include a sentence on recommendations.

Introduction:

1. When making reference of previous studies, there is no need to report estimates and their 95% confidence intervals.

2. A statement, .... “poor mental status has been identified as a key contributor to HIV epidemic among key populations,” requires a citation.

Results:

1. The statement, “the prevalence of psychological distress was statistically different in the three groups” lacks supporting statistical evidence to substantiate it.

2. Table 2:

a. All percentages should be row percentages to make sense of the relationships between variables

b. The p-value reported for the relationship between age and psychological distress need to be described whether it is a p-for trend or not. If it is not a p-value for trend, then there is also a need to mentions which comparison is being made.

Discussion:

1. When referencing previous studies, there is no need to report the estimates and their 95% confidence intervals.

2. Grammatically, use “a” and “an” as appropriately. FSW should be preceded by “an” and not “a”…similarly, it is “an MSM” and not “a MSM”

3. In the limitations of your study, you have pointed out some potential bias that could have been introduced in your study. However, you did not show or describe the potential impact of the introduced bias to your results.

4. One of the limitations you have mentioned is the cross-sectional nature of your study pointing that, you cannot elicit causality from this design. Does this mean the findings are of no use? Can you mention how can we make use of your findings despite the lack of causality?

5. The last paragraph of your discussion presents recommendations. I advise it be moved to an appropriate section.

Conclusion:

1. Your conclusion presents recommendations. I would advise you change the title of the section to Conclusion and Recommendation and then include concluding remarks as well as recommendations.

6. PLOS authors have the option to publish the peer review history of their article (what does this mean?). If published, this will include your full peer review and any attached files.

Reviewer #1: No

Reviewer #2: No

---

## [Author Response · Author response to Decision Letter 0]

7 Feb 2020

Response to reviewers

 Reviewer #1: Thanks for the opportunity to read and learn about your interesting work. May I offer the following suggestions.

Authors: We thank the reviewer for this encouraging comment. 

1. It is important to be clear about the direction of associations in key places like the Abstract and in the first paragraph of Discussion. I think most of what you say in the Results section of the Abstract is okay *except* regarding education. I recommend you end the list of factors with "hazardous alcohol consumption (aOR and 95% CI are listed here) were associated with PD." Then start a new sentence saying something like "People with secondary (aOR and 95% CI here) or higher (aOR and CI) education levels were LESS likely to report PD."

Then in the first paragraph of Discussion section (page 11), you should specify GREATER age and LOWER education level.

Authors: We thank the reviewer for these comments which have been taken into consideration in the abstract and discussion sections. 

HIV prevalence was 13.7% (95%CI = [12.2-15.2]). High age (25 years and over) [aOR = 1.24 (95% CI: 1.02–1.50)], being HIV positive [aOR = 1.80 (95% CI: 1.31–2.48)] and hazardous alcohol consumption [aOR = 1.52 (95% CI: 1.22–1.87)] were risks factors associated with PD. People with secondary [aOR = 0.52 (95% CI: 0.42-0.64)] or higher [aOR = 0.46 (95% CI: 0.32-0.64)] education levels were protective factors associated with PD. 

Discussion (page 10)

Factors associated with psychological distress other than HIV infection were greater age, lower education level and hazardous alcohol consumption. 

2. Where you refer to bivariate and multivariate methods, my understanding is these should be labeled Univariable and Multivariable (ending with -able instead of -ate). See https://www.ncbi.nlm.nih.gov/pmc/articles/PMC3679183/ for discussion.

Authors: We thank the reviewer for these comments. All the words have been relabeled (bivariate relabeled univariable and multivariate relabeled multivariable). 

Descriptive statistics, univariable and multivariable ordinal regression models were used for analysis.

For model building, characteristics that had a p-value <0.20 in univariable analysis were considered for the full multivariable models, which were subsequently finalized using a stepwise, backward elimination approach (p-value <0.05).

In univariable analysis, age, marital status, education level, religion, alcohol consumption, tobacco consumption, HIV infection status and the group of key population were associated with PD (Table 3). 

In multivariable analysis, after adjustment on the other variables, respondents who were 25 years old and older, were more likely to have severe/moderate or mild PD than respondents who were younger (p=0.026).

3. There are several places where you splice two sentences together with the word "and." Probably better to separate to 2 sentences, or use a semicolon instead of the "and." e.g., near bottom of page 2 after citations (11-13). 

Authors: We thank the reviewer for these comments which have been taken into consideration in the document. 

4. Specific:

Page 1 Abstract, last line in Methods section, should "model" have an -s on it ? Also, not sure if you can use abbreviation PD without introducing it, even though it appears later in the sentence in the name of the scale.

Authors: We thank the reviewer for these comments which have been taken into consideration in the document. 

Methods

Psychological distress (PD) was assessed with the Kessler Psychological Distress Scale. A blood sample was taken to test for HIV. Descriptive statistics, univariable and multivariable ordinal regression models were used for analysis.

5. Regarding Results section of Abstract, see note below about so many zeros after decimal points. for median and IQR, delete the ".0" because you are using only integer values. For the percentages, some of the ".0" are actually not correct. 19.0% should be 19.2% (392/2044). Although I did not calculate the CI's, those too are not correct and you will have to calculate those yourself (where it says 18.0-22.0 and 17.0-21.0).

Authors: We thank the reviewer for these comments which have been taken into consideration in the document. 

Results

The overall prevalence of mild PD among the three populations was 19.9% (95%CI= [18.3-21.8]) and was 19.2% (95%CI= [17.5-20.9]) for severe/moderate PD. HIV prevalence was 13.7% (95%CI = [12.2-15.2]). 

6. page 2, 2nd paragraph. I would change the 1st 2 sentences around a bit: "Some population subgroups, such as those at higher risk of HIV, are particularly burdened with mental health issues. According to UNAIDS (etc …)"

Authors: We thank the reviewer for these comments which have been taken into consideration in the document. 

Some population subgroups, such as those at higher risk of HIV, are particularly burdened with mental health issues. According to the UNAIDS, gay men and other men who have sex with men (MSM), female sex workers (FSW) and their clients, transgender people, people who inject drugs and prisoners and other incarcerated people are the main key population groups (7).

7. page 2, 8th line from bottom, insert "the" so it says "key contributor to the HIV epidemic"

page 3 line 7, add "of" so it says "prevalence of and factors associated with"

Authors: We thank the reviewer for these comments which have been taken into consideration in the document. 

Moreover, poor mental health has been identified as a key contributor to the HIV epidemic among key populations.

The aim of this study was to assess the prevalence of and factors associated with PD among MSM, FSW and DU in Togo in 2017. 

8. Need just a little more info on what "passives, actives" refers to - is it sexual position?

Authors: We thank the reviewer for these comments. These two notions refer to sexual positions: the active designates the one who penetrates and the passive, the penetrated.

9. Last paragraph on Kessler PD Scale - some of this text is repeated on the next page; delete one or the other.

Authors: We thank the reviewer for these comments which have been taken into consideration in the document. 

10. Page 4 -- should be only one L in syphilis. Then 2 lines down, delete s on "tests"

next paragraph is where Kessler PD repeat occurs. Also in that paragraph, change "allows to categorize" to "allows categorization of."

Authors: We thank the reviewer for these comments which have been taken into consideration in the document. 

Two HIV rapid tests were performed using SD Biolane Duo VIH/Syphilis rapid test kits. For each participant, a 4 mL blood sample was collected and a third test, ImmunoComb II VIH1-2 Comfirm (Orgenics Ltd, Israël) was used for confirmation. All biological test analyses were completed in the principal HIV laboratory research unit, the Molecular Biology Laboratory (CBMI) at the University of Lomé. 

Scores and operational definitions

Psychological Distress 

The score obtained from the scale allows categorization of participants into four categories of PD: severe (score ≥ 30), moderate (score: 25-29), mild (score: 20-24) and none (score < 20) (28).

11. page 5: along with "latent" mention that PO model is used for *ordinal* outcome variables.

Authors: We thank the reviewer for these comments which have been taken into consideration in the document. 

We considered Partial Proportional Odds model for the dependent variable Psychological Distress (PD). 

• The Proportional Odds (PO) model is appropriate when the response categories are based on one or more latent response variable. 

12. With backward elimination approach, what was the criterion for elimination? e.g., P<0.05, 0.01?

Authors: We thank the reviewer for these comments. The criterion for elimination was p<0.05 for the multivariable model. 

For model building, characteristics that had a p-value <0.20 in univariable analysis were considered for the full multivariable models, which were subsequently finalized using a stepwise, backward elimination approach (p-value <0.05).

13. page 6: first sentence regarding Kessler scores, leave off all the zeros after decimal for median and IQR scores, since these Kessler score values are always integers. On the other hand, in the next sentence some of the zeros after decimal point in the percentages are not correct. 20.0% is correct, but 61.0% should be 60.9%, and 19.0% should be 19.2%. You will need to recalculate your 95% CI's in this sentence and the rest of this paragraph - these should not all be ending with ".0%". 

Authors: We thank the reviewer for these comments which have been taken into consideration in the document. 

Prevalence of psychological distress 

The median Kessler score was 14 (IQR = [10-19]) for MSM, 18 (IQR = [13-23]) for FSW and 21 (IQR= [16-27]) for DU. About two-thirds (n=1,244; 61.0%) of the participants did not have psychological distress. The overall prevalence of mild psychological distress among the three populations was 19.9% (95%CI= [18.3-21.8]) and was 19.2% (95%CI= [17.5-20.9]) for the severe/moderate psychological distress. The prevalence of psychological distress was statistically different in the three groups (p-value <0.001). Among DU, the prevalence of mild psychological distress was 23.4%, 95%CI = [20-28] and 32.1%, 95%CI = [28-37] had severe/moderate psychological distress. Severe/moderate psychological distress was reported among one FSW in five (19.0%; 95%CI = [17-22]). The relationships between sociodemographic and clinical characteristics and psychological distress are presented in Table 2. 

14. I don't think you need Figure 1. Instead, just add the key population groups as another variable at the bottom of Table 2.

In Table 2, you need to make the same corrections to percentages (first row only) as in the text, namely change 61.0 to 60.9 and 19.0 to 19.2. On age, delete the ".0"s since all these numbers are integers only. Throughout the rest of Table 2, I think the ROW totals should add to 100% rather than the column totals. That helps the reader quickly approximate the odds ratios you will be presenting in Table 3.

Authors: We thank the reviewer for these comments which have been taken into consideration in the document. 

Table 2: Associations between psychological distressa and sociodemographic and clinical characteristics among key populations (N=2044)

 Psychological distress 

 No Mild Severe/Moderate 

P-value

Prevalence, n (%) 1244 (60.9) 408 (19.9) 392 (19.2) 

Age (years), median [IQR] 24 [21-30] 26 [21-33] 28 [22.5-34] <0.001*

Marital status, n (%) <0.001**

 Not married 1081 (62.7) 334 (19.4) 308 (17.9) 

 Married/Living with a partner 163 (50.8) 74 (23.0) 84 (26.2) 

Education level, n (%) <0.001**

 Primary school 292 (45.4) 164 (25.5) 187 (29.1) 

 Secondary school 725 (65.8) 208 (18.9) 169 (15.3) 

 College/University 227 (76.0) 36 (12.0) 36 (12.0) 

Religion, n (%) 0.003**

 Other/ Non-believer 136 (50.0) 67 (24.6) 69 (25.4) 

 Christian 973 (62.6) 297 (19.1) 284 (18.3) 

 Muslim 135 (61.9) 44 (20.2) 39 (17.9) 

Alcohol consumption, n (%) <0.001** 

 Non-drinker 502 (70.2) 117 (16.4) 96 (13.4) 

 Moderate drinking 244 (60.1) 83 (20.4) 79 (19.5) 

 Hazardous consumption 498 (54.0) 208 (22.5) 217 (23.5) 

Tobacco consumption, n (%) <0.001**

Non-smoker 1069 (63.3) 333 (19.7) 288 (17.0) 

Smoker 175 (49.4) 75 (21.2) 104 (29.4) 

HIV infection, n (%) 0.010**

No 1075 (86.4) 366 (89.7) 323 (82.4) 

Yes 169 (13.6) 42 (10.3) 69 (17.6) 

Key population <0.001** 

DU 200 (44.5) 105 (23.4) 144 (32.1) 

FSW 548 (57.6) 223 (23.4) 181 (19.0) 

MSM 496 (77.1) 80 (12.5) 67 (10.4) 

a Psychological distress was measured with the Kessler Psychological Distress Scale (10). Total Kessler score <20: No Psychological distress, 20-24: Mild disorder, 25-50: Moderate/Severe disorder. 

 * Kruskall-Wallis test; ** Chi-square test;

IQR: interquartile range

DU: drug user; FSW: female sex worker; MSM: men who have sex with me

15. Page 9: Change 1st sentence to: "RESULTS OF ordinal logistic regression models (etc...)" (add s to "models")

Line 2, here I would say univariable analysis instead of univariate. (and note you said "bivariate" in the abstract, which I think should also say "univariable")

Line 3, delete the word "factors" so it just says "were associated with …". Next paragraph, change multivariate to multivariable. Add an s so it says odds in "The odd of psychological" and then change "was" to "were"

Authors: We thank the reviewer for these comments which have been taken into consideration in the document. 

Factors associated with Psychological Distress

Results of ordinal logistic regression models among key population are presented in table 3. 

In univariable analysis, age, marital status, education level, religion, alcohol consumption, tobacco consumption, HIV infection status and the group of key population were associated with psychological distress (Table 3). 

In multivariable analysis, after adjustment on the other variables, respondents who were 25 years old and older, were more likely to have severe/moderate or mild psychological distress than respondents who were younger (p=0.026). Those with secondary school level or with college/university level were less likely to have severe/moderate or mild psychological distress compared to those who stopped in primary school (p<0.001). The odds of psychological distress were higher among hazardous drinkers (aOR=1.52; 95%CI= [1.22-1.87]) compared to non-drinkers. HIV serological status was a factor associated with psychological distress (aOR= 1.80; 95%CI [1.31-2.48]). Compared to DU, FSW (aOR= 0.55; 95%CI = [0.43-0.68]) and MSM (aOR= 0.33; 95%CI = [0.24-0.44]) were less likely to have severe/moderate or mild psychological distress.

16. Table 3: change column headers to Univariable and Multivariable. I think you should explain in a footnote that for nearly all variables, the left 2 columns are the same, and the right 2 columns are the same, because the assumption of proportional odds was met for those variables and therefore you used the same value for both columns. (only the tobacco smoker and HIV infection OR's are different.)

Authors: We thank the reviewer for these comments which have been taken into consideration in the document. 

Table 3: Factors associated with psychological distress among key population in Togo (Partial proportional Odds model) #.

 (Psychological distress: 1 = Severe/Moderate, 2 = Mild, 3 = No)

 Univariable Multivariable

 1 vs (2 and 3) (1 and 2) vs 3 P-value 1 vs (2 and 3) (1 and 2) vs 3 P-value

 OR (95 % CI) OR (95 % CI) aOR (95 % CI) aOR (95 % CI) 

Age (years) 

< 25 Reference <0.001 Reference 0.026 

≥ 25 1.77 (1.48-2.11) 1.77 (1.48-2.11) 1.24 (1.02-1.50) 1.24 (1.02-1.50) 

Marital status <0.001 -

Not married Reference - - 

Married/Living with a partner 1.63 (1.29-2.04) 1.63 (1.29-2.04) - - 

Education level <0.001 <0.001

Primary school Reference Reference 

Secondary 0.44 (0.36-0.52) 0.44 (0.36-0.52) 0.52 (0.42-0.64) 0.52 (0.42-0.64) 

College/University 0.27 (0.20-0.37) 0.27 (0.20-0.37) 0.46 (0.32-0.64) 0.46 (0.32-0.64) 

Religion <0.001 -

Other/ Non-believer Reference - - 

Christian 0.62 (0.48-0.78) 0.62 (0.48-0.78) - - 

Muslim 0.63 (0.44-0.88) 0.63 (0.44-0.88) - - 

Alcohol consumption <0.001 <0.001

Non-drinker Reference Reference 

Moderate drinking 1.56 (1.22-2.0) 1.56 (1.22-2.0) 1.28 (0.98-1.65) 1.28 (0.98-1.65) 

Hazardous consumption 2.0 (1.64-2.45) 2.0 (1.64-2.45) 1.52 (1.22-1.87) 1.52 (1.22-1.87) 

Tobacco consumption <0.001 -

Non-smoker Reference 

Smoker 1.84 (1.46-2.32) 1.48 (1.22-1.80) - - 

HIV infection 0.004 <0.001

No Reference Reference 

Yes 1.46 (1.08-1.96) 1.02 (0.79-1.32) 1.80 (1.31-2.48) 1.25 (0.94-1.65) 

Key population <0.001 <0.001 

DU Reference Reference 

FSW 0.56 (0.45-0.69) 0.56 (0.45-0.69) 0.55 (0.43-0.68) 0.55 (0.43-0.68) 

MSM 0.23 (0.18-0.30) 0.23 (0.18-0.30) 0.33 (0.24-0.44) 0.33 (0.24-0.44) 

aOR: adjusted Odds ratio; OR: Odds ratio

# For nearly all variables, the left 2 columns are the same, so are the right 2 columns, because the assumption of proportional odds was met for those variables and therefore we used the same value for both columns. (only the tobacco smoker and HIV infection OR's are different) 

17. page 11, it is confusing to use "respectively" when there are 2 different lists. I suggest "Among MSM, FSW, and DU in Togo, the overall prevalence of mild and severe/moderate psychological distress was 20.0% and 19.2% respectively."

Authors: We thank the reviewer for these comments which have been taken into consideration in the document. 

Discussion

Among MSM, FSW, and DU in Togo, the overall prevalence of mild and severe/moderate psychological distress was 19.9% and 19.2% respectively. 

18. 2nd paragraph, could you say more about whether and why these percentages reporting PD in other countries are higher or lower than in your study? In the sentence where you cite the 23.9% in South Africa, you could say "compared to 39.2% in our study."

Authors: We thank the reviewer for these comments. There is a gap between the percentages of PD in other countries and in our study mainly because of the age differences. Indeed, the population considered in our study is relatively young (median age = 25 years) unlike in other studies. 

Zaller N, Yang C, Operario D, Latkin C, McKirnan D, O’Donnell L, et al. Alcohol and cocaine use among Latino and African American MSM in 6 US cities. Journal of substance abuse treatment. 2017;80:26. 

Gyawali B, Choulagai BP, Paneru DP, Ahmad M, Leppin A, Kallestrup P. Prevalence and correlates of psychological distress symptoms among patients with substance use disorders in drug rehabilitation centers in urban Nepal: a cross-sectional study. BMC psychiatry. 2016;16(1):314.

Best wishes with the manuscript and your research.

Authors: We thank the reviewer for this encouraging comment and for all the inputs to enhance the quality of the paper. 

 

Reviewer #2: Review for article titled: Prevalence and Factors Associated with Psychological Distress among Key Populations in Togo, 2017. Overall, the paper reads well, congratulations!

I have minor comments that I think need to be addressed:

Authors: We thank the reviewer for this encouraging comment. 

Abstract

1. The write up has several grammatical errors that should be worked on.

Authors: We thank the reviewer for these comments which have been taken into consideration in the document. 

2. Use of abbreviations should come after you have spelled the long form earlier. In the abstract, the author has used the abbreviation PD for the first time before it is written in full.

Authors: We thank the reviewer for these comments which have been taken into consideration in the document. 

Abstract

Objectives

Mental health is a largely neglected issue among in Sub-Saharan Africa, especially among key populations. The aim of this study was to estimate the prevalence of psychological distress (PD) and to assess the factors associated among males who have sex with males (MSM), female sex workers (FSW) and drug users (DU) in Togo in 2017. 

3. In reporting factors associated with PD, the author could choose to report such factors in groups; that is those that increase the risk and those that reduce the risk.

Authors: We thank the reviewer for these comments which have been taken into consideration in the document. 

Abstract

Results

High age (25 years and over) [aOR = 1.24 (95% CI: 1.02–1.50)], being HIV positive [aOR = 1.80 (95% CI: 1.31–2.48)] and hazardous alcohol consumption [aOR = 1.52 (95% CI: 1.22–1.87)] were risks factors associated with PD. People with secondary [aOR = 0.52 (95% CI: 0.42-0.64)] or higher [aOR = 0.46 (95% CI: 0.32-0.64)] education levels were protective factors associated with PD.

Results

Factors associated with Psychological Distress

In multivariable analysis, after adjustment on the other variables, respondents who were 25 years old and older, were more likely to have severe/moderate or mild PD than respondents who were younger (p=0.026). HIV serological status was a risk factor associated with PD (aOR= 1.80; 95%CI [1.31-2.48]). The odds of PD were higher among hazardous drinkers (aOR=1.52; 95%CI= [1.22-1.87]) compared to non-drinkers. Those with secondary school level or with college/university level were less likely to have severe/moderate or mild PD compared to those who stopped in primary school (p<0.001).

4. The conclusion sounds more like a recommendation. I advise that you give concluding statements based on your findings. You may want to title the section as Conclusion and Recommendation and include a sentence on recommendations.

Authors: We thank the reviewer for these comments which have been taken into consideration in the document. 

Conclusion and Recommendations

This is the first study conducted among a large, nationally representative sample of Key Population in Togo. The prevalence of PD is high among Key Population in Togo and was associated to HIV infection. The present study indicates that mental health care must be integrated within health programs in Togo with a special focus to key populations through interventions such as social support groups. 

Introduction:

1. When making reference of previous studies, there is no need to report estimates and their 95% confidence intervals.

Authors: We thank the reviewer for these comments which have been taken into consideration in the document. 

2. A statement, .... “poor mental status has been identified as a key contributor to HIV epidemic among key populations,” requires a citation.

Authors: We thank the reviewer for the comment. The statement above and the sentence which follows it in the text, have the same citation (11–13). 

Moreover, poor mental health has been identified as a key contributor to the HIV epidemic among key populations. Indeed, MSM who are mentally depressed engage in unprotected sex and substance abuse (11–13). Several studies have demonstrated a significant association between depressive symptoms and HIV and other sexually transmitted infections (STI) among MSM (14–16).

Results:

1. The statement, “the prevalence of psychological distress was statistically different in the three groups” lacks supporting statistical evidence to substantiate it.

Authors: We thank the reviewer for the comment. We have put the p-value of the chi-square test below figure 1 but have put it now in the text since we removed the figure 1. 

The prevalence of PD was statistically different in the three groups (p-value <0.001).

2. Table 2:

a. All percentages should be row percentages to make sense of the relationships between variables

b. The p-value reported for the relationship between age and psychological distress need to be described whether it is a p-for trend or not. If it is not a p-value for trend, then there is also a need to mentions which comparison is being made.

Authors: We thank the reviewer for these comments. All percentages have been changed into row percentages in Table 2. The p-value reported for the relationship between age and psychological distress is for a Kruskall-Wallis test. 

Discussion:

1. When referencing previous studies, there is no need to report the estimates and their 95% confidence intervals.

Authors: We thank the reviewer for these comments which have been taken into consideration in the revised document. 

2. Grammatically, use “a” and “an” as appropriately. FSW should be preceded by “an” and not “a” …similarly, it is “an MSM” and not “a MSM”

Authors: We thank the reviewer for these comments which have been taken into consideration in the document. 

Our study has some limitations. First, we worked on three populations, but it is possible that an FSW may be an DU or an MSM may also be an DU. 

3. In the limitations of your study, you have pointed out some potential bias that could have been introduced in your study. However, you did not show or describe the potential impact of the introduced bias to your results.

Authors: We thank the reviewer for these comments which have been taken into consideration in the document. 

Since our study was based on self-reporting, memory bias and social desirability bias cannot be ruled out. These bias might have had an effect on the prevalence rates of PD. Also, as we conducted a cross-sectional study, causality between the factors identified and psychological distress could not be addressed.

4. One of the limitations you have mentioned is the cross-sectional nature of your study pointing that, you cannot elicit causality from this design. Does this mean the findings are of no use? Can you mention how can we make use of your findings despite the lack of causality?

Authors: We thank the reviewer for these comments. A cross sectional studies cannot prove causality. 

Like any cross-sectional study, our study cannot conclude a causality between the factors identified and psychological distress. However our results are useful to the extent that, acting on the associated factors will reduce the risk of the event occurring.

5. The last paragraph of your discussion presents recommendations. I advise it be moved to an appropriate section.

Authors: We thank the reviewer for these comments which have been taken into consideration in the document. The paragraph has been moved to the conclusion. 

Conclusion:

1. Your conclusion presents recommendations. I would advise you change the title of the section to Conclusion and Recommendation and then include concluding remarks as well as recommendations.

Authors: We thank the reviewer for these comments. In the revised version of the manuscript, we changed the title of the section to Conclusion and recommendations. 

Conclusion and Recommendations 

This is the first study conducted among a large, nationally representative sample of key populations in Togo. The prevalence of PD is high among Key Population in Togo and was associated to HIV infection. Mental health care such as social support groups must be integrated within health and prevention programs dedicated to key populations in Togo in order to promote a holistic perspective of their health. In order to confirm the high prevalence of PD, studies in the general population or comparative studies are needed. Also, since PD a multifactorial process, it would be useful to couple a qualitative study with a quantitative study.

---

## [Decision Letter · Decision Letter 1]

26 Mar 2020

PONE-D-19-29894R1

Prevalence and factors associated with psychological distress among key populations in Togo, 2017.

PLOS ONE

Dear MPH TCHANKONI,

Thank you for submitting your manuscript to PLOS ONE. After careful consideration, we feel that it has merit but does not fully meet PLOS ONE’s publication criteria as it currently stands. Therefore, we invite you to submit a revised version of the manuscript that addresses the points raised during the review process.

We would appreciate receiving your revised manuscript by May 10 2020 11:59PM. To enhance the reproducibility of your results, we recommend that if applicable you deposit your laboratory protocols in protocols.io, where a protocol can be assigned its own identifier (DOI) such that it can be cited independently in the future. For instructions see: http://journals.plos.org/plosone/s/submission-guidelines#loc-laboratory-protocols

We look forward to receiving your revised manuscript.

Kind regards,

Joel Msafiri Francis, MD, MS, PhD

Academic Editor

PLOS ONE

Additional Editor Comments (if provided):

Thank you for addressing all the previous comments. Please kindly address the few additional comments from the reviewers.

Reviewers' comments:

Reviewer's Responses to Questions

**Comments to the Author**

1. If the authors have adequately addressed your comments raised in a previous round of review and you feel that this manuscript is now acceptable for publication, you may indicate that here to bypass the “Comments to the Author” section, enter your conflict of interest statement in the “Confidential to Editor” section, and submit your "Accept" recommendation.

Reviewer #1: (No Response)

Reviewer #2: All comments have been addressed

2. Is the manuscript technically sound, and do the data support the conclusions?

Reviewer #1: Yes

Reviewer #2: Yes

3. Has the statistical analysis been performed appropriately and rigorously? 

Reviewer #1: Yes

Reviewer #2: Yes

4. Have the authors made all data underlying the findings in their manuscript fully available?

Reviewer #1: Yes

Reviewer #2: Yes

5. Is the manuscript presented in an intelligible fashion and written in standard English?

Reviewer #1: Yes

Reviewer #2: Yes

6. Review Comments to the Author

Reviewer #1: Thank you for addressing my comments. There are still a few small mistakes, and some comments that are addressed in your response to me but I am not sure if they are addressed in the manuscript.

Abstract line 37: add "at risk for HIV" after "key populations."

line 56 delete s on "risks" so it will say "risk factors"

line 57: delete "People with" (the people themselves are not the protective factors)

line 304: change "these bias" to "these biases"

A few items that you answered to me but I still think need to be addressed in the manuscript itself (thanks for numbering the comments, that is a good idea). :

Comment 8: change "passives, actives" to "passive or active sexual position"

Comment 11: the ordinal nature of the outcome variable is what actually matters to the proportional odds model. You could have multiple latent variables involved that led to a multi-categorical outcome variable (e.g., anxious extroverts, anxious introverts, non-anxious extroverts, non-anxious introverts) that was not ordinal, and in that case the proportional odds model would not be correct.

Comment 18: The information that you provided to me regarding age differences is helpful, but I think that info should be provided a little more specifically in the manuscript in line 266.

Two additional item related to comment 11:

11.1: Did you actually report anything based on the Partial Proportional Odds model (PPO) as opposed to the Proportional Odds model (PO) ? If not I suggest you delete the description of PPO in methods in lines 173-178.

11.2: add the word "ordinal" in line 155 so it says "four ordinal categories"

Congrats on the great work!

Reviewer #2: Credits to the authors for addressing all the comments. I have minor additional comments for consideration before publication in the attached document. Thanks for making an effort to contribute to science as we work to end the HIV pandemic.

7. PLOS authors have the option to publish the peer review history of their article (what does this mean?). If published, this will include your full peer review and any attached files.

Reviewer #1: No

Reviewer #2: No

---

## [Author Response · Author response to Decision Letter 1]

27 Mar 2020

Response to reviewers

 Reviewer #1: Thank you for addressing my comments. There are still a few small mistakes, and some comments that are addressed in your response to me but I am not sure if they are addressed in the manuscript.

Authors: We thank the reviewer for this encouraging comment. 

1. Abstract line 37: add "at risk for HIV" after "key populations."

line 56 delete s on "risks" so it will say "risk factors" 

line 57: delete "People with" (the people themselves are not the protective factors) 

line 304: change "these bias" to "these biases"

Authors: We thank the reviewer for these comments which have been taken into consideration in the abstract and discussion sections. We have reworded the sentence before the one referring to biases and therefore “bias” is in the singular. 

Abstract

Objectives

line 37: Mental health is a largely neglected issue among in Sub-Saharan Africa, especially among key populations at risk for HIV.

line 56 and 57: (…) and hazardous alcohol consumption [aOR = 1.52 (95% CI: 1.22–1.87)] were risk factors for PD. Secondary [aOR = 0.52 (95% CI: 0.42-0.64)] or higher [aOR = 0.46 (95% CI: 0.32-0.64)] education levels were protective factors associated with PD. 

Discussion

line 304: Since our study was based on self-reporting, social desirability bias cannot be ruled out. This bias could have underestimated the prevalence rates of PD. 

2. Comment 8: change "passives, actives" to "passive or active sexual position"

Authors: We thank the reviewer for these comments which have been taken into consideration in the document. 

Each seed from the first wave selected had to represent at least one sub-group: the actives (the ones who penetrate), the passives (the penetrated), bisexuals, gays, etc. 

3. Comment 11: the ordinal nature of the outcome variable is what actually matters to the proportional odds model. You could have multiple latent variables involved that led to a multi-categorical outcome variable (e.g., anxious extroverts, anxious introverts, non-anxious extroverts, non-anxious introverts) that was not ordinal, and in that case the proportional odds model would not be correct.

Authors: We thank the reviewer for these comments which have been taken into consideration in the document and the sentence has been removed from the section. 

Statistical analysis

We considered Partial Proportional Odds model for the dependent variable PD. Descriptive statistics were performed and results were presented with frequency tabulations and percentages. Prevalence rates were estimated with their 95% confidence interval. For model building, characteristics that had a p-value <0.20 in univariable analysis were considered for the full multivariable models, which were subsequently finalized using a stepwise, backward elimination approach (p-value <0.05). Predictor variables were selected as those found to be relevant according to the literature review. All computations were conducted using SAS version 9.4 (SAS Institute Inc., Cary, NC, USA). 

4. Comment 18: The information that you provided to me regarding age differences is helpful, but I think that info should be provided a little more specifically in the manuscript in line 266.

Authors: We thank the reviewer for these comments which have been taken into consideration in the document. 

Discussion

(…) This prevalence is similar to the overall moderate/severe psychological distress prevalence in our study among the three key populations (19.0%). However, this estimate was higher than that reported among MSM in our study (10.4%). There is a gap between the percentages of PD in other countries and in our study mainly because of the age differences. Indeed, the population considered in our study is relatively young (median age = 25 years) unlike in other studies (35,36). Also, different tools have been used in the other studies to measure PD as the Patient Health Questionnaire (PHQ) (19), the Center for Epidemiologic Studies Depression Scale (CES-D) (40) and K-6 (35). 

5. 11.1: Did you actually report anything based on the Partial Proportional Odds model (PPO) as opposed to the Proportional Odds model (PO)? If not I suggest you delete the description of PPO in methods in lines 173-178.

Authors: We thank the reviewer for these comments which have been taken into consideration in the document and the description has been removed from the section. 

Statistical analysis

We considered Partial Proportional Odds model for the dependent variable PD. Descriptive statistics were performed and results were presented with frequency tabulations and percentages. Prevalence rates were estimated with their 95% confidence interval. For model building, characteristics that had a p-value <0.20 in univariable analysis were considered for the full multivariable models, which were subsequently finalized using a stepwise, backward elimination approach (p-value <0.05). Predictor variables were selected as those found to be relevant according to the literature review. All computations were conducted using SAS version 9.4 (SAS Institute Inc., Cary, NC, USA). 

6. 11.2: add the word "ordinal" in line 155 so it says "four ordinal categories"

Psychological Distress 

The K10 has been examined and validated among several populations and aims at measuring anxiety and depression with a 10-item questionnaire, each question pertaining to an emotional state and a five-level response scale for each response. The score obtained from the scale allows categorization of participants into four ordinal categories of PD: severe (score ≥ 30), moderate (score: 25-29), mild (score: 20-24) and none (score < 20) (28).

Congrats on the great work!

Authors: We thank the reviewer for this encouraging comment and for all the inputs to enhance the quality of the paper. 

 

Reviewer #2: Credits to the authors for addressing all the comments. Thanks for making an effort to contribute to science as we work to end the HIV pandemic.

Authors: We thank the reviewer for this encouraging comment. 

1. Abstract: In the sub-section results line 56/57, the statement, “….were risks factors associated with PD.”, should be changed to “…were risk factors for PD” or “….were factors associated with PD.”

Authors: We thank the reviewer for these comments which have been taken into consideration in the document. 

Abstract

Results

Line 56/57: (…) were risk factors for PD. Secondary [aOR = 0.52 (95% CI: 0.42-0.64)] or higher [aOR = 0.46 (95% CI: 0.32-0.64)] education levels were protective factors associated with PD.

2. Results section: 

Line 231: the statement, “HIV serological status was a risk factor associated with PD…” should read, “HIV positive serological status was a risk factor for PD…” Having it as just HIV serological status is ambiguous. 

Authors: We thank the reviewer for these comments which have been taken into consideration in the document. 

Main document

Results

In multivariable analysis, after adjustment on the other variables, respondents who were 25 years old and older, were more likely to have severe/moderate or mild PD than respondents who were younger (p=0.026). HIV positive serological status was a risk factor for PD (aOR= 1.80; 95%CI [1.31-2.48]).

3. Line 302: “…. FSW may be an DU or an MSM may also be an DU.” Should change to “…. FSW may be a DU or an MSM may also be a DU”

Authors: We thank the reviewer for these comments which have been taken into consideration in the document. 

Discussion

(…) Our study has some limitations. First, we worked on three populations, but it is possible that an FSW may be a DU or an MSM may also be a DU. However, to classify the population, we took into account only the entry point.

4. Line 304/305: “These bias might have had an effect on the prevalence rates of PD.” This statement first needs to be qualified; will the biases overestimate or underestimate the prevalence? To answer this question, you will need to dissect each bias separately. Consider revision appropriately. 

Authors: We thank the reviewer for these comments which have been taken into consideration in the document. 

However, to classify the population, we took into account only the entry point. Since our study was based on self-reporting, social desirability bias cannot be ruled out. This bias could have underestimated the prevalence rates of PD.

---

## [Editor Report · Decision Letter 2]

31 Mar 2020

Prevalence and factors associated with psychological distress among key populations in Togo, 2017.

PONE-D-19-29894R2

Dear Dr. TCHANKONI,

We are pleased to inform you that your manuscript has been judged scientifically suitable for publication and will be formally accepted for publication once it complies with all outstanding technical requirements.

With kind regards,

Joel Msafiri Francis, MD, MS, PhD

Academic Editor

PLOS ONE

Additional Editor Comments (optional):

I think would helpful to consider revising the description of active and passive sexual activity - I think in a sexual encounter both people are active. Would you perhaps consider using the " Top (for active)" and " Bottom (for passive)"

.
---

## [Editor Report · Acceptance letter]

6 Apr 2020

PONE-D-19-29894R2 

Prevalence and factors associated with psychological distress among key populations in Togo, 2017. 

Dear Dr. TCHANKONI:

I am pleased to inform you that your manuscript has been deemed suitable for publication in PLOS ONE. Congratulations! Your manuscript is now with our production department. 

With kind regards,

on behalf of

Dr. Joel Msafiri Francis 

Academic Editor

PLOS ONE